# ATP128 Clinical Therapeutic Cancer Vaccine Activates NF-κB and IRF3 Pathways through TLR4 and TLR2 in Human Monocytes and Dendritic Cells

**DOI:** 10.3390/cancers14205134

**Published:** 2022-10-20

**Authors:** Roberta Pascolutti, Lakshmi Yeturu, Géraldine Philippin, Stéphane Costa Borges, Magali Dejob, Marie-Laure Santiago-Raber, Madiha Derouazi

**Affiliations:** 1AMAL Therapeutics, 1205 Geneva, Switzerland; 2Boehringer-Ingelheim GmbH, 55216 Ingelheim, Germany

**Keywords:** therapeutic cancer vaccine, self-adjuvanticity, toll-like receptor 2, toll-like receptor 4, dendritic cells activation

## Abstract

**Simple Summary:**

Peptide-based cancer vaccines require adjuvants to increase their immunogenicity to mount an antigen-specific reliable immune response. We have previously shown that immunization with self-adjuvanting peptide-based vaccines issued from our KISIMA vaccine platform, and targeting tumor-specific antigens, resulted in potent immune responses in preclinical models. The aim of our study was to assess the molecular mechanism of action of the first human candidate vaccine derived from the KISIMA platform, ATP128. By using an in vitro (THP-1 cell line) and ex vivo model system (human monocyte-derived dendritic cells), we were able to demonstrate that ATP128 relies on both the cell-penetrating peptide and TLR agonist domain present in the construct to activate the NF-κB and IRF3 pathways, in a self-adjuvanting manner. Importantly, we demonstrated that ATP128 is the first therapeutic vaccine able to activate human DCs through TLR2 and TLR4.

**Abstract:**

The use of cancer vaccines is a promising therapeutic strategy able to stimulate anti-tumor immunity by inducing both humoral and cellular immunity. In this study, antigen presenting cells play a key role by inducing a strong activation of the T cell-mediated adaptive immune response, essential for the anti-tumor potential of cancer vaccines. The first human candidate vaccine created from the KISIMA platform, ATP128, bears three tumor-associated antigens highly expressed in colorectal cancer tissues. At the N-terminus, the cell-penetrating peptide allows the antigen delivery inside the cell and, together with the TLR agonist-derived peptide at the C-terminus, ensures the activation of the monocyte-derived dendritic cells. Here, we show that ATP128 leads to both NF-κB and IRF3 pathway activation, with subsequent pro-inflammatory cytokines and type I Interferon release, as well as an increase in the expression of costimulatory molecules, alongside an upregulation of MHC class I molecules. This cellular immune response involves TLR2 and TLR4, for both membrane and intracellular signaling. We demonstrated an endocytic component in ATP128’s activity by combining the use of a variant of ATP128 lacking the cell-penetrating peptide with endocytosis inhibitors. Importantly, this internalization step is detemined essential for the activation of the IRF3 pathway. This study validates the design of the self-adjuvanting ATP128 vaccine for cancer immunotherapy.

## 1. Introduction

One of the different approaches exploited in cancer treatment implies the use of therapeutic cancer vaccines, which rely on the ability of the patient’s immune system to elicit an immune response specifically attacking and killing cancer cells, in an antigen-specific manner, sparing the healthy cells. The antitumor immune response following the therapeutic cancer vaccine often appears ineffective and of low amplitude. A recent strategy to overcome this issue has been demonstrated through a modular self-adjuvanting cancer vaccine KISIMA^®^ platform [1]. The technology of the KISIMA platform is based on the assembly within a single chimeric fusion protein of three elements essential to generate potent antitumoral cellular immunity: a proprietary cell-penetrating peptide (CPP) for antigen delivery, a proprietary Toll-like receptor (TLR) agonist (TLRag) peptide with self-adjuvant properties, and a modular multi-antigenic domain (Mad), which can be tailored specifically to any given tumor type. The combination of the CPP, Mad, and TLRag simultaneously activate antigen-presenting cells (APCs) like cross-presenting dendritic cells (DC) and their antigen cross-presentation, essential for eliciting a multiantigenic immune response. Importantly, cell-penetrating peptides increase antigenic delivery by conferring the ability to cross the plasma membrane, allowing antigens to be processed and presented on the Major Histocompatibility Complex (MHC) class I and class II, subsequently inducing CD8 and CD4 T-cell activation, respectively [2]. Murine forms of KISIMA derived vaccines demonstrated a marked efficacy in animal models, alone or in combination with an immune check-point inhibitor (anti-PD-1) [1].

The first human candidate vaccine created from the KISIMA platform, ATP128, is one such chimeric recombinant protein vaccine in which the Mad consists of three antigens selected because of their overexpression in a large percentage of Colorectal Cancer (CRC) tissues: carcinoembryonic antigen (CEA), Survivin, and Achaete-scute complex homolog 2 (ASCL2) [3]. ATP128 is currently in a Phase 1b clinical trial (KISIMA-01) testing for the treatment of patients with stage IV microsatellite stable/mismatch repair proficient (MSS/MMRp) CRC (NCT04046445). Here, we show that ATP128 is the first therapeutic vaccine, to our knowledge, capable of activating both TLR2 and TLR4 leading to activation of both MyD88 (Myeloid differentiation primary response 88) and TRIF (TIR-containing adapter-inducing interferon-β)-mediated pathways in a self-adjuvanting manner.

The TLR agonist domain in KISIMA aims to enhance vaccine potency and works as an adjuvant by delivering the target antigen to professional APCs such as DCs, thereby leading to their activation [4]. The maturation of DCs provided by TLR agonists induces an increase in the expression of costimulatory molecules and production of cytokines, alongside enhanced MHC class I processing machinery [5], translating into higher antigen presentation and T-cell activation, essential for a powerful and long-lasting immunity.

TLRs are transmembrane proteins that can be classified according to their cellular localization: TLR1, 2, 4, 5, 6 and 10 are expressed at the cell surface and recognize the molecular patterns of extracellular microbes [6]; TLR3, 7, 8 and 9 are localized to the membranes of intracellular organelles, such as endosomes [7] and are activated by nucleic acid-based molecules. The self-adjuvanting activity exploited by the KISIMA platform is based on an Annexin II-derived peptide (Anaxa) and is a TLR2 agonist when monomeric [8] and a TLR4 agonist when tetrameric [9]. TLR2 specificity is driven by its interaction with other TLRs, heterodimerizing with either TLR1 or TLR6, thereby behaving differently from the homomeric TLR4 [10]. Upon binding with their cognate ligands, TLR2 and TLR4 both activate the adaptor protein MyD88 via their cytoplasmic tail; after internalization, TLR4 can also activate the TRIF pathway. Both pathways lead to the downstream activation of nuclear factor κB (NF-κB), activating protein-1 (AP-1) and/or IRF3 (Interferon Regulatory Factor 3), ultimately resulting in the transcription of pro-inflammatory cytokines, chemokines, type I interferons, and immunomodulators [11]. Their different localization patterns, responses to diverse stimuli, and activation of multiple pathways, indicate that TLR agonists are important tools able to propagate various functions within the immune response, and can be utilized as adjuvants in infectious disease and cancer vaccines [11].

Endocytosis is a known cellular uptake mechanism used by various pharmacological substances. Endocytic pathways can be broadly classified as clathrin-dependent (CDE) and clathrin-independent (CIE) endocytosis. Lipid rafts are an example of CIE, and are microdomains enriched in cholesterol, glycosphingolipids, and unsaturated fats, which laterally transport molecules across the plasma membrane. Several proteins are known to prefer this route into cells as it is energetically favorable [12]. As such, it is thought that the CPP derived from EBV-ZEBRA potentiates its entry into cells through a lipid-raft mediated mechanism involving endocytosis [13].

Here, we describe the involvement of TLR2 and TLR4 as receptors through which the human vaccine ATP128 signals to activate the cellular immune response. In an ex vivo system, ATP128 relies on the activation of both the surface and internalized TLR4 to activate human dendritic cells, leading to cytokine secretion and increased expression of costimulatory molecules, with a signature similar to the one of MPLA, a known TLR4 agonist used as prophylactic vaccine adjuvant in the clinic [14]. Moreover, we have identified a lipid raft-mediated pathway that contributes to the intracellular activation, given by the presence of the CPP.

By the mechanism of self-adjuvanticity, ATP128 will allow DCs activation, antigenic presentation to tumor specific CD4 and CD8 T cells as observed with KISIMA surrogate treatment in tumor mouse models [1].

## 2. Materials and Methods

### 2.1. Cell Lines

THP-1 Dual™, and THP-1-Dual™ KO-TLR2, KO-TLR4 cells were purchased from InvivoGen (San Diego, CA, USA). They were cultured in RPMI 1640 (1×) Glutamax with 25 mM HEPES (Gibco), penicillin, and streptomycin (100×; Gibco, Waltham, MA, USA), 10% heat-inactivated fetal bovine serum (FBS; Gibco) and 50 μg/mL Normocin (InvivoGen).

### 2.2. Vaccines

The ATP128 vaccine construct was designed in-house and produced in *E. coli* by Boehringer-Ingelheim (ATP128 ENG DP) through a 3-step purification process, wherein two hydrophobic steps are separated by a cation exchange. ATP128 *w*/*o* CPP A2 and ATP128 *w*/*o* Anaxa A2 were designed and produced in-house through HIS-tag purification. Vaccines were prepared by dilution in vaccine buffer. Endotoxin content was quantified in each vaccine batch using a LAL chromogenic assay (Appendix A) (ToxinSensorTM Chromogenic LAL Endotoxin Assay Kit, Genscript, Piscataway, NJ, USA, L00350). Only the batches with an endotoxin level less than or equal to 10 EU/mg protein (according to guidelines) were used for further experiments (Appendix A).

### 2.3. Antibodies

For THP-1 activation assays: anti-mouse TLR2 (mab2-mtlr2) and anti-human TLR4 (mabg-htlr4) antibodies were purchased from InvivoGen. For flow cytometry: CD83 FITC (BD #556910), CD80 PE (BD #557227), CD86 APC (BD #555660), CD45 BV786 (BD #563716), HLA-ABC Pe-Cy7 (BD #561349), HLA-DR BV421 (BD #564244), and CD11c APC (BD #559877) were purchased from BD Biosciences. To determine cell viability, LIVE/DEAD Fixable Yellow Stain (LDY #L34968, Invitrogen, Waltham, MA, USA) was used.

### 2.4. For Western Blot

Anti-β-actin (ab8227), anti-phospho IRF3 S386 (ab76493), and anti-IRF3 (ab68481) antibodies were obtained from Abcam. Anti-phospho NF-κB S536 (93H1; #3033) and anti-NF-κB (L8F6; was #6956S) antibodies were purchased from Cell Signaling Technologies; and the anti-GAPDH (#39-8600) antibody from Invitrogen.

### 2.5. TLR Activation Experiment

Dose Response Assays: THP-1 Dual cells were seeded in 96-well TPP tissue culture plates (Sigma-Aldrich, St. Louis, MO, USA) and then stimulated with increasing concentration of ATP128 and its truncated vaccine forms. After 18 h, supernatants were recovered and incubated with either QUANTI-Blue™ (InvivoGen) or QUANTI-Luc™ (InvivoGen) to assess secreted embryonic alkaline phosphatase (SEAP) levels, as a measure of NF-κB activity or the luciferase reporter gene (Lucia, InvivoGen) for IRF3 activity, respectively. SEAP release was quantified by a spectrophotometer at 620 nm while Lucia activity was assessed by luminescence read at 100 ms-1.

### 2.6. Blocking Antibodies Experiments

THP-1 Dual™ and THP-1-Dual™ KO-TLR2, KO-TLR4 cells were pre-treated for 1 h with blocking antibodies (1 μg/mL) alone or in combination and/or with CLI-095 (760 nM). Cells were then stimulated with 300 nM ATP128 vaccine forms and incubated overnight (37 °C, 5% CO_2_); then NF-κB and IRF3 activation were assessed as previously described.

### 2.7. Endocytosis Inhibition Experiments

THP-1 cells were pre-treated with blocking antibodies and/or CLI-095, in presence or not of 5 µg/mL chlorpromazine (CPZ; Sigma-Aldrich) or 10 µg/mL chloroquine (CHQ; Sigma-Aldrich). After a 1 h incubation, the cells were stimulated with 300 nM ATP128 FL or ATP128 *w*/*o* CPP, incubated overnight at 37 °C, then assessed by a TLR activation assay as described above.

### 2.8. NF-κB and IRF3 Phosphorylation

For kinetics studies, THP-1 cells were treated with 300 nM ATP128 constructs at different time points. The assay was blocked by placing the plate at 4 °C. Cells were lysed with lysis buffer (RIPA buffer; Cell Signaling, Danvers, MA, USA). For studies involving stimulation with endocytosis inhibitors, THP-1 cells were pre-treated for 1 h with 5 µg/mL CPZ, 10 µg/mL CHQ, or 40 µM dynasore (DYN: Sigma-Aldrich), or for 45 min with 40 µM Pitstop (PIT; Sigma-Aldrich) or 10 µM methyl-β-cyclodextrin (MBCD; Sigma-Aldrich). After treatment incubation, the cells were stimulated with 300 nM ATP128 for an additional hour.

### 2.9. Cell Preparation

Monocytes were isolated from buffy coats of normal blood donors. Briefly, PBMCs were isolated on a Ficoll density gradient (GE Healthcare, Boston, MA, USA) and resuspended into complete RPMI medium. Monocytes were then isolated by aggregation at cold temperature (two cycles) to remove contaminating T cells. Another Ficoll density gradient was then performed to isolate enriched monocytes, which were resuspended and maintained in complete RPMI medium supplemented with 1000 U/mL GM-CSF (Miltenyi Biotech, North Rhine-Westphalia, Germany) and 800 U/mL IL-4 (Miltenyi Biotech) for five days, after which the medium and cytokines were renewed. After renewal, the cells were cultured for an additional 4 days, then used for experiments. The cells were always maintained in a humidified environment at 37 °C with 5% CO_2_.

### 2.10. Ex Vivo Human moDCs Activation

Surface staining: Human moDCs plated in 24-well plates were loaded with the different variants of human vaccine ATP128, in the presence or absence of blocking antibodies anti-TLR2 or anti-TLR4; small molecule inhibitor, CLI-095; 600 nM MPLA (InvivoGen); or 1 ng/mL FSL-1 (InvivoGen). Plates were incubated over 24 h at 37 °C; after cell scraping and washes, and the cells were stained for the monitoring of CD80, CD83, CD86, HLA-ABC, and HLA-DR expression by flow cytometry. Gating strategy is shown in Appendix A.

### 2.11. Cytokine Signatures

Human moDCs plated in 24-well plates were loaded with ATP128 or its truncated forms with or without blocking antibodies anti-TLR2 or anti-TLR4; small molecule inhibitor CLI-095; 600 nM MPLA (InvivoGen); or 1 ng/mL FSL-1 (InvivoGen). Plates were incubated over 6 h at 37 °C; sample supernatants were collected, and the production of cytokines was analyzed using a commercial multiplex assay (Inflammation 20-Plex Human ProcartaPlexTM Panel) according to the manufacturer’s instructions.

### 2.12. Statistics

Statistical analyses were performed using GraphPad Prism software (one-way ANOVA test or unpaired *t* test) and considered statistically significant if *p* < 0.1 for TLR activation experiment or *p* < 0.05 for the moDCs activation and cytokines signature.

## 3. Results

### 3.1. CPP and TLRag Domains of ATP128 Contribute to Both NF-κB and IRF3 Activation Pathways

To understand the contribution to downstream signaling of each domain present within the KISIMA vaccine, truncated forms of ATP128 were engineered, with either the CPP or the TLRag (Anaxa) domain removed (Figure 1A). The ability of these vaccine constructs to activate the NF-κB and IRF3 pathways in a human monocytic cell line (THP-1 Dual) through a dose response assay was analyzed. The absence of the Anaxa domain affected the maximal activation of the NF-κB pathway (Figure 1B, left panel), resulting in a significant reduction of EC50 compared with the full length (FL) ATP128 (8.2 nM versus 69.7 nM, respectively). The construct without (*w*/*o*) CPP displayed a low effect on NF-κB activation (EC50 27 nM). Notably, at a high concentration of the vaccine (100 nM), the absence of CPP showed only a mild decrease in activation when compared to the FL construct, while at low concentration (10 nM) it drastically affected NF-κB activation, a decrease that is further enhanced in absence of the Anaxa domain. This suggests that the CPP component of the vaccine minimally contributes to the activation of the NF-κB pathway at higher doses. In contrast, both the N- and C-terminus of ATP128 (CPP and Anaxa domains) appeared to be essential for IRF3 activation as their absence in the vaccine prevented 80 to 90% of IRF3 activation, respectively, in comparison with the FL construct (Figure 1B, right panel).

### 3.2. ATP128 Activates NF-κB and IRF3 Pathways Engaging Both TLR2 and TLR4

The contribution of the surface receptors upon ATP128 stimulation was investigated by analyzing NF-κB and IRF3 activation in the presence or absence of specific anti-TLR2 and anti-TLR4 blocking antibodies (Figure 1C). A significant decrease of NF-κB activation is observed when surface TLR2 (30%) but not surface TLR4 is blocked. However, when both these receptors are simultaneously blocked, this effect is more pronounced with a 53% decrease in NF-κB activation (Figure 1C, left panel). Regarding IRF3, a slight decrease is noticed when anti-TLR2 is used, while upon anti-TLR4 treatment there is significant inhibition of approximately 60% in this pathway’s activation. Additionally, the combination of the two surface TLR2 and TLR4 blockade leads to a drastic and significant impairment of this pathway (Figure 1C, right panel). To further assess the involvement of an intracellular TLR2 and TLR4 signaling component, TLR2 and TLR4 knocked-out cell lines were used. As depicted in Figure 1D, the complete knockout of TLR2 resulted in a significant defect in both pathways when compared with the control. A decrease of 31% was observed for NF-κB activation while one of 44% was observed for IRF3 activation when compared with the control, thereby suggesting a signaling-specific role for endocytosed TLR2. Contrarily, the abrogation of TLR4 did not impair the activation of the NF-κB pathway but its deficiency led to a strong decrease in the IRF3 pathway (90%). Together, these data suggest that in an in vitro system, both TLR2 and TLR4, play an essential role in the IRF3 pathway but TLR2 appears to be the primary receptor for NF-κB activation.

To further investigate the specific contribution of intracellular TLR4, a selective inhibitor of TLR4 (CLI-095/TAK-242) [15] that specifically suppresses TLR4 signaling was used. As shown in Figure 1E, left panel, the inhibition of both surface and endocytosed TLR4 by CLI-095 in THP-1 cells leads to only a minor decrease in the NF-κB pathway. However, the combination of the TLR4 inhibitor with anti-TLR2 results in a highly significant decrease of this pathway. The importance of TLR4 is further highlighted by the complete impairment of IRF3 activation upon the selective inhibition of TLR4 alone or in combination with TLR2. These results are confirmed with the use of TLR2 and TLR4 knocked-out cells (Figure 1F), suggesting that both the activity of TLR2 and TLR4 is essential to properly induce the activation of the NF-κB and IRF3 pathways.

### 3.3. Impact of Endocytosis on NF-κB and IRF3 Activaton

To investigate the involvement of the endocytic pathway leading to NF-κB activation by ATP128, endocytosis inhibitors, chlorpromazine (CPZ), an inhibitor of clathrin-mediated endocytosis, and chloroquine (CHQ), an endosomal maturation inhibitor, were used. Blocking endocytosis alone had a limited impact on the activation of the NF-κB pathway (Figure 2A); however, it greatly affected IRF3 activation (Figure 2B). The simultaneous inhibition of both surface receptors, TLR2 and TLR4, and endocytosis lead to significant impairment in the activation of both the NF-κB and IRF3 pathways (Figure 2A,B), potentially suggesting the importance of signaling at both extra- and intra-cellular levels. To address the remaining activation of these pathways, we investigated the effect of blocking endocytosis upon stimulation with ATP128 *w*/*o* CPP. In this context, we evaluated the contribution of the CPP portion of the ATP128 construct for its ability to enter cells through a possible lipid-raft mediated mechanism as reported for zebra, allowing protein transduction via direct translocation through the lipid bilayer [13]. As shown in Figure 2C, the absence of CPP completely abrogates the remaining NF-κB activation, thereby confirming the importance of this domain in the absence of available TLR2 and TLR4 receptors at the cell surface and within the endosomes. The IRF3 pathway was further inhibited when cells were stimulated with ATP128 *w*/*o* CPP compared to ATP128 FL (Figure 2D), also corroborating the importance of this domain in this pathway.

### 3.4. ATP128 Induces IRF3 Phosphorylation with Delayed Kinetics Compared to NF-κB

Subsequently, early steps of activation of both NF-κB and IRF3 pathways were investigated. The activation of these pathways was assessed through a kinetic of NF-κB and IRF3 phosphorylation. As shown in Figure 3A, the phosphorylation of NF-κB was observed from 15 min to 1 h, decreasing thereafter; however, IRF3 follows a delayed phosphorylation kinetic beginning at 30 min and peaking at 2h. In addition, two experiments with different kinetics were performed (separately, 0, 30 min, 1 h, 2 h and 4 h or 0, 1 h, 2 h and 4 h), showing a comparable increase in IRF3 phosphorylation at 2 h. Similarly, the contribution of the CPP and the Anaxa domains of ATP128 in the early activation of NF-κB was assessed through a kinetics experiment. As shown in Appendix A, the ATP128 FL construct leads to a maximal NF-κB phosphorylation at 1 h, while ATP128 *w*/*o* Anaxa is unable to induce any activation. Conversely, the absence of CPP only decreases the extent of phosphorylation at early time points but peaks in the FL construct, suggesting a minor contribution of this domain in the early activation of the NF-κB pathway. To evaluate the contribution of endocytosis in ATP128 signaling, as shown in Figure 2, different endocytic inhibitors; either targeting the early steps at the plasma membrane (such as dynasore DYN, chlorpromazine CPZ or pitstop PIT), the endosomal maturation (like chloroquine CHQ), or lipid-raft inhibitors (e.g., β-methylcyclodextrin MBCD); were tested. The different screened endocytosis/trafficking inhibitors confirmed that the activation of NF-κB is partially inhibited upon blocking of endocytosis (Figure 3B). Notably, cells treated with either chlorpromazine, pitstop, or chloroquine show an impairment in NF-κB phosphorylation/activation, as previously demonstrated through the THP-1 activation assay (Figure 2A).

### 3.5. TLR4 Is Essential to Activated Human moDCs by ATP128

After having assessed the involvement of TLR2 and TLR4 in an in vitro system, the ATP128 activation pattern was also measured in human monocyte-derived dendritic cells (moDCs). Monocytes were isolated from different healthy donors (buffy coats) and differentiated in vitro into moDCs. The upregulation of the costimulatory molecules, CD80 and CD86, and the maturation marker CD83 [16] at the cell surface of moDCs was observed when cells were stimulated for 24 h with ATP128 (Figure 4A). The upregulation was not affected when extracellular TLR2 was blocked, but drastically decreased when both extra- and intracellular TLR4 signaling were inhibited with CLI-095. Similarly, the upregulation of the HLA class I molecule was affected in the presence of the TLR4 inhibitor (CLI-095); however, no statistically significant decrease was observed for the HLA class II molecule (Figure 4B). Moreover, the observed upregulation of co-stimulatory molecules was comparable to the one induced by MPLA (Figure 5A), a well-known TLR4 agonist, further confirming a major involvement of TLR4 in the activation of dendritic cells, the main targeted cells of the KISIMA platform. However, no statistically significant upregulation was observed for HLA class II molecules upon ATP128 stimulation (Figure 5B). It is worth noting that the two model systems investigated show a different expression level of the TLRs at the cell surface, with lower expression of TLR4 in the THP-1 cell line when compared with its expression level of moDCs (Appendix A).

Cytokines secreted by human moDCs were assessed upon stimulation with the KISIMA vaccine. After a 6 h stimulation of moDCs with ATP128 in the presence or absence of blocking antibodies, supernatant was collected, and pro- and anti-inflammatory cytokine expressions were quantified (Figure 6A). The stimulation of moDCs with ATP128 led to the expression of IL-6 and TNF-α, known to be released through the activation of the NF-κB, together with IP-10, induced upon IRF3 activation, as well as IL-10, whose release is after the activation of both pathways [11,17,18]. This cytokine expression panel was only slightly affected when the surface receptors were blocked (α-TLR2 or α-TLR4); however, it was greatly impaired upon the complete inhibition of TLR4 (CLI-095) alone or in combination with anti-TLR2 (CLI-095 + α-TLR2). These data further confirm that ATP128 activates the target cells mainly through TLR4, via both MyD88 and TRIF-mediated pathways. Notably, TLR4 inhibition also affected type I IFN secretion, consecutive to IRF3 activation preferentially [19], confirming the role of ATP128 in activating both NF-κB and IRF3 pathways. The secretion of IL-12, a major marker associated to dendritic cells activation, was measured. The expression of cytokines upon activation with MPLA (TLR4 agonist) or FSL-1 (TLR2 agonist) was compared to ATP128 stimulation. As shown in Figure 6B, ATP128 induced the expression of IL-6, IP-10, IL-12 and type I interferons at a similar level to MPLA and to a higher extent compared with FSL-1, further confirming the involvement of both TLR4 and TLR2 by ATP128, with TLR4 being predominant.

### 3.6. CPP and Anaxa Work Synergestically to Activate Human moDCs

The ability of ATP128 and its truncated forms to activate human DCs was also investigated. As shown in Figure 7A, both CPP and Anaxa are required for moDCs activation, as indicated by upregulation of CD80, CD86, and CD83, or HLA class I molecules (Figure 7B). Moreover, the combination of these truncated versions of ATP128 with blocking antibodies specific for TLR2 and TLR4 totally impaired the residual activation as seen in the potency assay (Figure 2C,D). In parallel, the cytokine profile of the DCs upon stimulation with ATP128 and its different truncated forms was further assessed. As shown in Figure 8, TNF-α, IL-6, and IL-10 levels partially decreased in the absence of CPP in the KISIMA construct. Importantly, the truncation of Anaxa led to a major decrease in the level of all cytokines including IL-12. Notably, IP-10 (IRF3 activation), as well as IFN-β expression levels, were totally impaired when either CPP or Anaxa were not present in the construct, confirming that both the domains are required for TRIF-mediated cytokines secretion. Overall, the addition of blocking TLR2 and TLR4 further decreased the secretion of IL-6, TNF-α, and INF-α. All these data confirm the importance of the role of both the CPP and TLR agonist in proficiently activating human dendritic cells.

## 4. Discussion

Studies have previously demonstrated that KISIMA platform-derived vaccines elicit an efficient multiantigenic T-cell response due to their CPP and Anaxa domains [1]. The Anaxa domain is an Annexin II-derived peptide that can trigger the activation of both TLR2 [8] and TLR4 [9], conferring self-adjuvanticity and enhancing the immune response.

Ten TLRs have been described so far in humans, which are involved in innate and adaptive immune responses. Binding to natural pathogen ligands causes the TLR proteins to dimerize and activate NF-κB, and other transcription factors such as IRF3, which promotes the upregulation of several genes including those encoding pro-inflammatory cytokines and costimulatory molecules. Different studies have shown that the synergistic involvement of multiple TLRs leads to a better APC activation [20,21,22,23], by enhancing the expression of costimulatory molecules and secretion of proinflammatory cytokines. This translates into an increased activation of T cells in vitro [20]. Also, it has been previously shown that MyD88- associated TLRs synergize with TRIF-associated TLRs for the induction of several pro-inflammatory cytokines, IFN-β production, and NF-κB nuclear translocation [24]. Here, we demonstrate that the human ATP128 vaccine is able to strongly induce APCs via concomitant signaling through TLR2 and both intracellular and surface TLR4, with slightly different receptor engagement in the two model systems investigated. Notably, our data show that in an in vitro system (THP-1 cell line), ATP128 signals mostly through TLR2 in activating the NF-κB pathway, while ex vivo (human moDC) this role is played by TLR4. The difference between the data obtained in an in vitro system and ex vivo could be explained by the different expression levels of TLRs observed at the cell surface, with a lower surface TLR4 expression in the THP-1 model system, leading to a different requirement of these receptors for ATP128 signaling. By using specific TLR2 and TLR4-genetically deficient cell lines, it is reasonable to hypothesize an intracellular TLR2 signaling pathway, a scenario described in a recently reported study, showing that Staphylococcus aureus infection of human monocytes activates a TLR2-dependent endosomal signaling pathway, leading to type I IFN induction in a TRIF-independent manner [25]. Through which pathway ATP128 would lead to TLR2 internalization and type I IFN release will be the purpose of future investigations. On the contrary, in the in vitro settings of the present study, the IRF3 pathway is mostly activated through surface and intracellular TLR4. These data suggest that TLR2 and TLR4 can compensate for the absence of each other in activating the NF-κB pathway, as they target similar genes.

Importantly, there are two types of NF-κB activation downstream of TLR4 signaling: a MyD88-dependent pathway crucial for the release of inflammatory cytokines such as IL-1β, TNF-α, and IL-6; and a MyD88-independent/TRIF-dependent pathway resulting in the induction of IP-10 [26]. The former induces an early-phase activation of NF-κB, whereas the latter leads to a late-phase activation of the transcription factor, as shown by a delayed NF-κB activation in MyD88^−/−^ cells [27]. Using specific inhibitors acting at the plasma membrane and endosomal level, we demonstrated that, at least in vitro, the intracellular pathway is neither required nor essential for the activation of the NF-κB pathway; however, its contribution is substantial in inducing the IRF3 pathway, recapitulating LPS signaling [26]. The contribution of the late phase could be dissected by using blocking antibodies in combination with ATP128 *w*/*o* CPP stimulation, confirming the requirement of CPP for protein internalization [13] and activation of the intracellular/TRIF-mediated pathway. This is consistent with a recent report demonstrating the ability of the CPP domain to deliver vaccine inside the cells [28]; however, the CPP domain is not required for activating the NF-κB pathway. The TRIF-mediated pathway is activated only intracellularly, and it plays a pivotal role only when the surface signaling is impaired. Finally, this internalization step might be the basis for the delay in IRF3 activation compared with NF-κB, as similarly demonstrated in previous studies [29,30]. Notably, the activation of moDCs by ATP128 through surface receptors induces pro-inflammatory cytokines like IL-6 and TNF-α, known to enhance the adaptive immune response by stimulating APC maturation, inhibiting regulatory T cells, and suppressing tolerance [31].

Studies performed in vitro with the THP-1 monocytic cell line surprisingly show that in absence of the Anaxa, ATP128 still retains its ability to activate the NF-κB pathway but not IRF3, a topic that requires further investigations. Noteworthy, the KISIMA vaccine ATP125 shares a similar moDC activation pattern [1] despite containing different antigens in its multiantigenic domain, suggesting that the Mad domain is not involved in TLR engagement. On the contrary, our ex vivo studies show that ATP128 is fully functional when both the CPP and the Anaxa domains are present, highlighting both the essential role of the Anaxa domain and its synergy with the CPP. As previously shown, the CPP domain is essential in delivering the vaccine inside the cells, allowing an MHC class I presentation of the antigens [28]; here, we show that it additionally impacts the activation of moDC by enabling the signaling through intracellular TLR. In the present study, we also demonstrate the absolute requirement of both the CPP and Anaxa domains in activating the IRF3 pathway both in vitro and ex vivo, essential for the induction of type I interferons, a first step in anti-tumoral T-cell functionality. Notably, IRF3 is required in APCs for optimal T-cell effector function [32]. Some early reports showed the ability of type I IFN to boost HLA class-I expression and simultaneous tumor antigens presentation to CD8 T, leading to a better immune recognition and cytotoxic effect on cancer cells [33,34].

Given their feature to link the innate with the adaptive immune response, TLR agonists are novel adjuvants to enhance DC function for efficient T-cell priming in cancer vaccine therapies. The most developed TLR4-stimulating adjuvant is monophosphoryl lipid (MPL), the first to obtain approval for use in human vaccines against viral pathogens such as herpes zoster, Hepatitis B, and human papillomavirus (HPV) [35]. Stimulation of TLR4 through MPL induces NF-κB activation with subsequent secretion of the pro-inflammatory cytokines TNF-α and IL-6, leading to APCs maturation [36], similarly to what we observed with the KISIMA vaccine, ATP128, in studies with human DCs. The pro-inflammatory cytokines and type I IFN induced upon ATP128 stimulation would then create an environment favorable to complete the activation of CD4 and CD8 lymphocytes toward their effector functions during an antitumor response.

The KISIMA vaccine ATP128 is to our best knowledge, the first therapeutic vaccine able to activate human DCs through TLR2 and TLR4 binding in a self-adjuvanting manner, exploiting both MyD88- and TRIF-mediated pathways. This leads to an increased APC activation that shall lead to subsequent improvement and amplification of tumor-specific T-cell responses. This assumption is supported by results from the ongoing KISIMA-01 clinical trial, wherein 50% of the evaluated patients treated with ATP128, a cellular immune response against at least one out of three antigens was observed, as determined by IFN-γ ELISpot analyses of patient PBMCs after the third vaccination [3]. This is an essential requirement for tumor cell recognition and killing, supporting the use of ATP128 in colorectal patients in the ongoing KISIMA-01 clinical trial (NCT04046445).

## 5. Conclusions

In conclusion, we show that the human ATP128 peptide-based vaccine is able to activate DCs through TLR2 and TLR4 signaling in a self-adjuvanting manner, inducing both MyD88- and TRIF-mediated pathways.

## Figures and Tables

**Figure 1 cancers-14-05134-f001:**
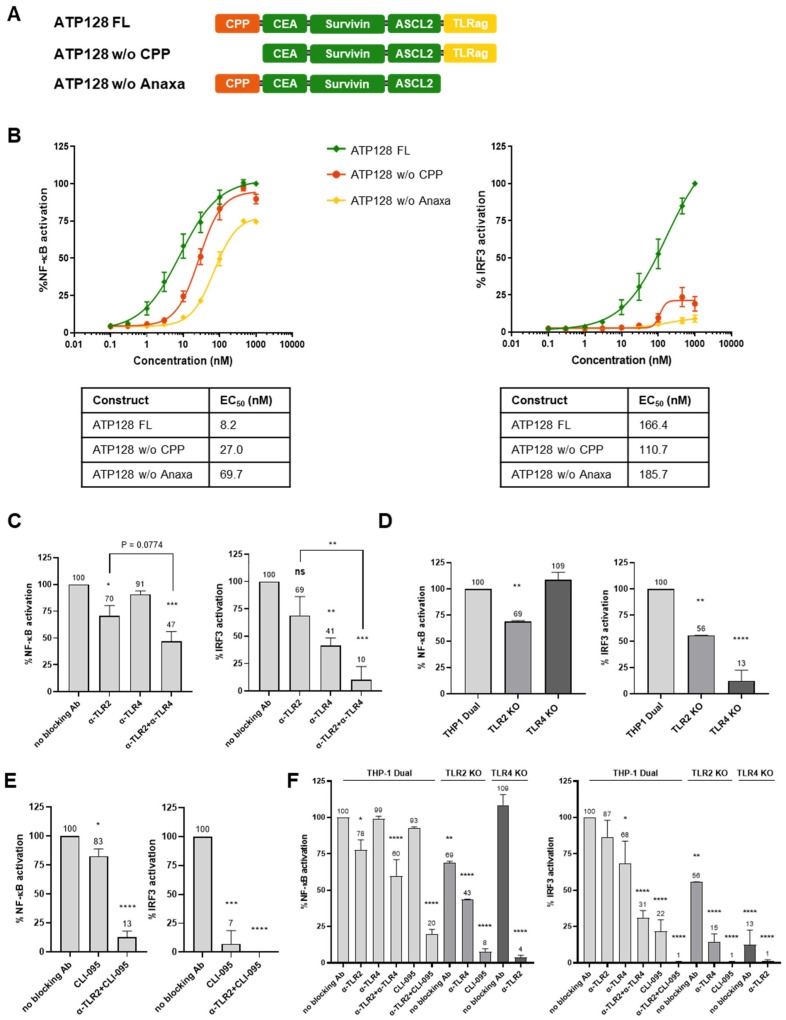
Functionality of KISIMA vaccine. (**A**) Scheme of ATP128 constructs. ATP128 full length (FL) is comprised of a Cell Penetrating Peptide (CPP) at the N-terminus, a Multiantigenic domain (Mad), and a TLR agonist at the C-terminus (Anaxa). (**B**) THP-1 Dual cells were incubated with increasing concentrations of ATP128 vaccine constructs. After 18 h, supernatant was recovered and either SEAP activity was measured by QUANTI-Blue assay (left panel), or Luciferase activity was measured by QUANTI-Luc assay (right panel). The EC50s of the different constructs were calculated from the obtained dose-response curve using the GraphPad Prism software. Experiments shown here represent the average of % of activation of 3 biological replicates. (**C**) THP-1 Dual cells were incubated with 300 nM ATP128 in presence or not of blocking antibodies anti-TLR. After 18 h, cell supernatants were recovered, assessed, and represented as described above. The values reported here are the mean of 4 biological replicates. Values were compared using a one-way ANOVA test to the untreated sample (positive control—no blocking antibody). (**D**) THP-1 Dual cells wild-type (WT) or knocked-out (KO) for specific TLR were stimulated with 300 nM ATP128. After 18 h, the cell supernatants were recovered, assessed, and represented as described above. The values reported here are a mean of 3 biological replicates. Values were compared via one-way ANOVA test to the WT sample. (**E**) THP-1 Dual cells were incubated with 300 nM ATP128 in presence or not of the TLR4 inhibitor CLI-095. After 18 h, the cell supernatants were recovered, assessed, and represented as described above. The values reported here are a mean of 4 biological replicates. Values were compared via one-way ANOVA test to the untreated sample (positive control). (**F**) THP-1 Dual WT, and TLR2 KO and TLR4 KO cells were incubated with 300 nM ATP128 in presence or not of antibodies anti-TLR and CLI-095. After 18 h, the cell supernatants were recovered, assessed, and represented as described above. The values reported here are a mean of 3 biological replicates. Values were compared via a one-way ANOVA test to the untreated sample in WT cells. **** *p*-value < 0.0001, *** *p*-value < 0.001, ** *p*-value < 0.01, * *p*-value < 0.1. ns: no significance.

**Figure 2 cancers-14-05134-f002:**
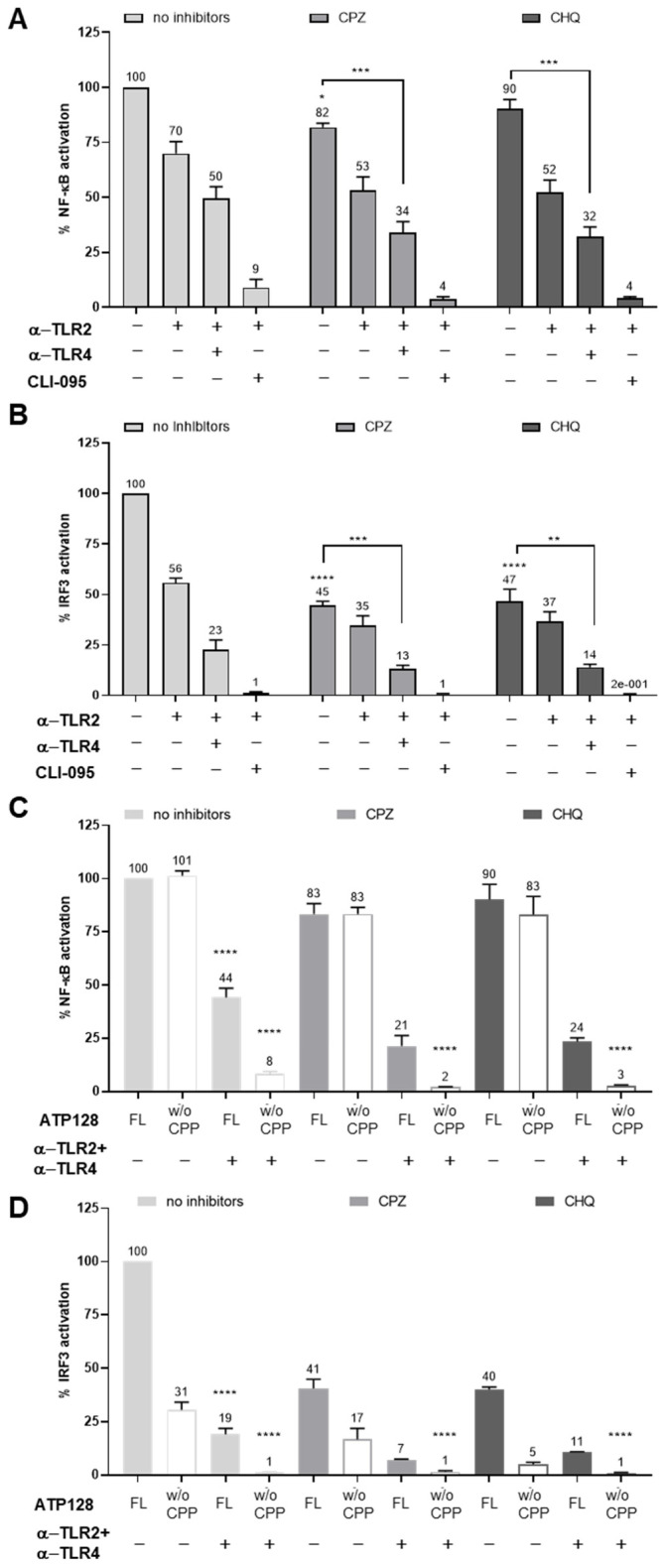
Characterization of the intracellular NF-κB pathway. (**A**,**B**) THP-1 Dual cells were pretreated with endocytosis inhibitors, such as chlorpromazine (CPZ) to inhibit clathrin-mediated endocytosis or chloroquine (CHQ), an inhibitor of endosomal maturation, in the presence or not of anti-TLR antibodies or of the inhibitor CLI-095 for one hour. Cells were subsequently stimulated with 300 nM ATP128. After 18 h, supernatant was recovered and either (**A**) SEAP activity was measured by QUANTI-Blue assay, or (**B**) Luciferase activity was measured by QUANTI-Luc assay. The values here reported are a mean of 3 biological replicates. Values were compared via either a one-way ANOVA test to the untreated control sample or unpaired t-test to the control of the group (CPZ or CHQ). (**C**,**D**) THP-1 Dual cells were pretreated with chlorpromazine (CPZ) or chloroquine (CHQ) in the presence or not of antibodies anti-TLR2/4 for one hour. Cells were subsequently stimulated with 300 nM ATP128 FL or ATP128 *w*/*o* CPP. After 18 h, supernatant was recovered and either (**C**) SEAP activity was measured by QUANTI-Blue assay, or (**D**) Luciferase activity was measured by QUANTI-Luc assay. Filled-in bars represent stimulation with ATP128 FL, while the outlines represent the stimulation with ATP128 *w*/*o* CPP. The values here reported are a mean of 3 biological replicates. Values were compared via a one-way ANOVA test to the untreated control sample stimulated with ATP128 FL. **** *p*-value < 0.0001, *** *p*-value < 0.001, ** *p*-value < 0.01, * *p*-value < 0.1.

**Figure 3 cancers-14-05134-f003:**
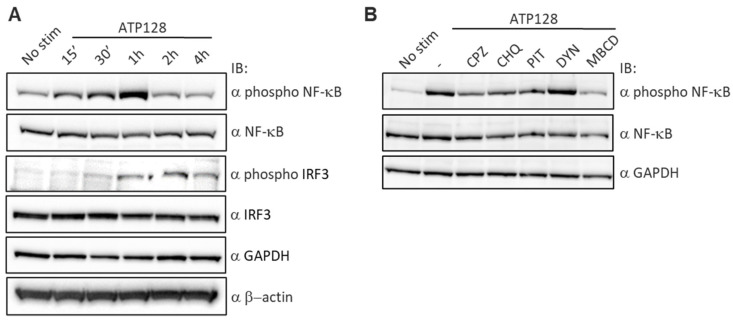
Extent of NF-κB and IRF3 phosphorylation upon ATP128 stimulation. (**A**) THP-1 Dual cells were stimulated with 300 nM ATP128 at different time points and the phosphorylation of NF-κB and IRF3 was assessed through Western blot using specific phospho-antibodies. Anti-β actin and anti-GAPDH were used as normalizers. (**B**) THP-1 Dual cells were pretreated with different endocytosis inhibitors for 1 h and then stimulated with 300 nM ATP128 for an additional hour; the phosphorylation of NF-κB was assessed by Western blot, as described above. The uncropped blots are shown in Appendix A.

**Figure 4 cancers-14-05134-f004:**
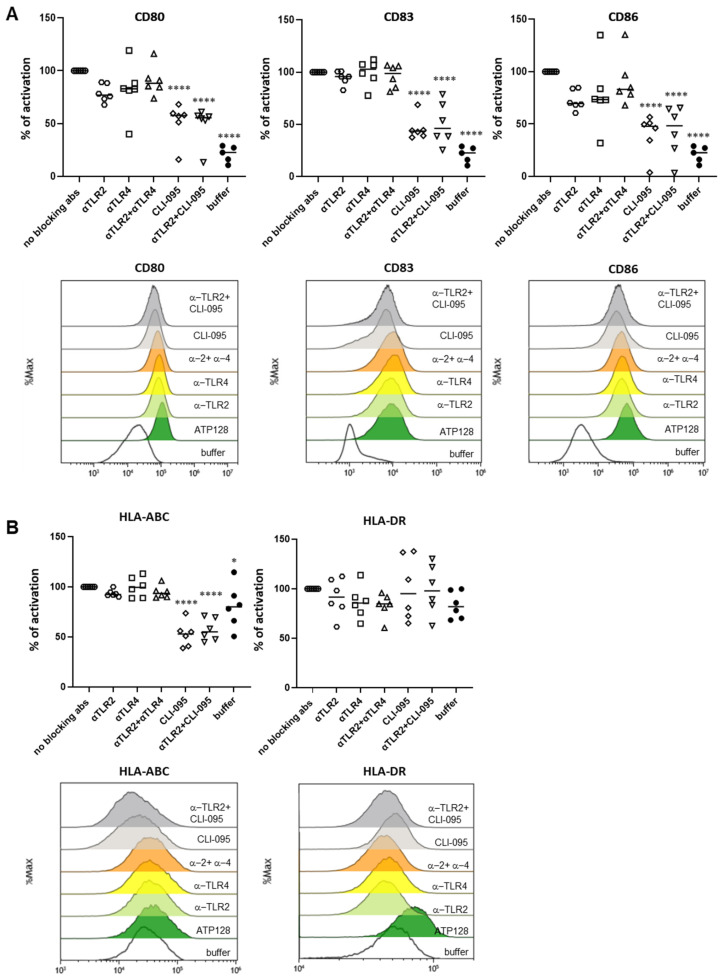
ATP128 engages mostly TLR4 for human moDC activation in an ex vivo system. (**A**,**B**) Human moDC were pre-treated with anti-TLR2, anti-TLR4 or CLI-095 for 1 h, then stimulated for 24 h with 300 nM ATP128. Cells were collected after stimulation, and surface costimulatory molecules (**A**) and HLA class I and II molecules (**B**) were analyzed through flow cytometry. Each shape is representative of a different buffy coat. Histograms of one representative buffy coat are shown below. Values were compared via one-way ANOVA test to the sample stimulated with ATP128 FL. **** *p*-value < 0.0001, * *p*-value < 0.05.

**Figure 5 cancers-14-05134-f005:**
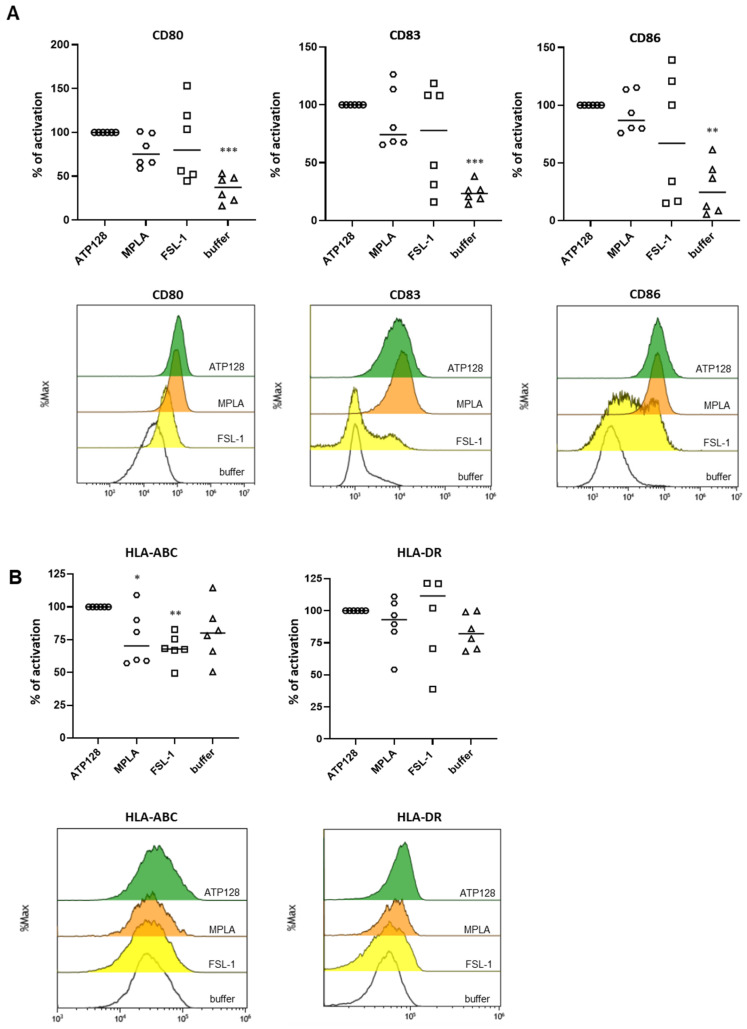
ATP128 activates human moDCs in a profile similar to the TLR4 agonist, MPLA, in an ex vivo system. (**A**,**B**) Human moDC were incubated for 24 h with 300 nM ATP128 or 600 nM MPLA or 1 ng/mL FSL-1. Cells were collected after stimulation and surface costimulatory molecules (**A**) and HLA class I and class II molecules (**B**) were analyzed through flow cytometry. Each shape is representative of a different buffy coat. Histograms of one representative buffy coat are shown below. Values were compared via one-way ANOVA test to the sample stimulated with ATP128 FL., *** *p*-value < 0.001, ** *p*-value < 0.01, * *p*-value < 0.05.

**Figure 6 cancers-14-05134-f006:**
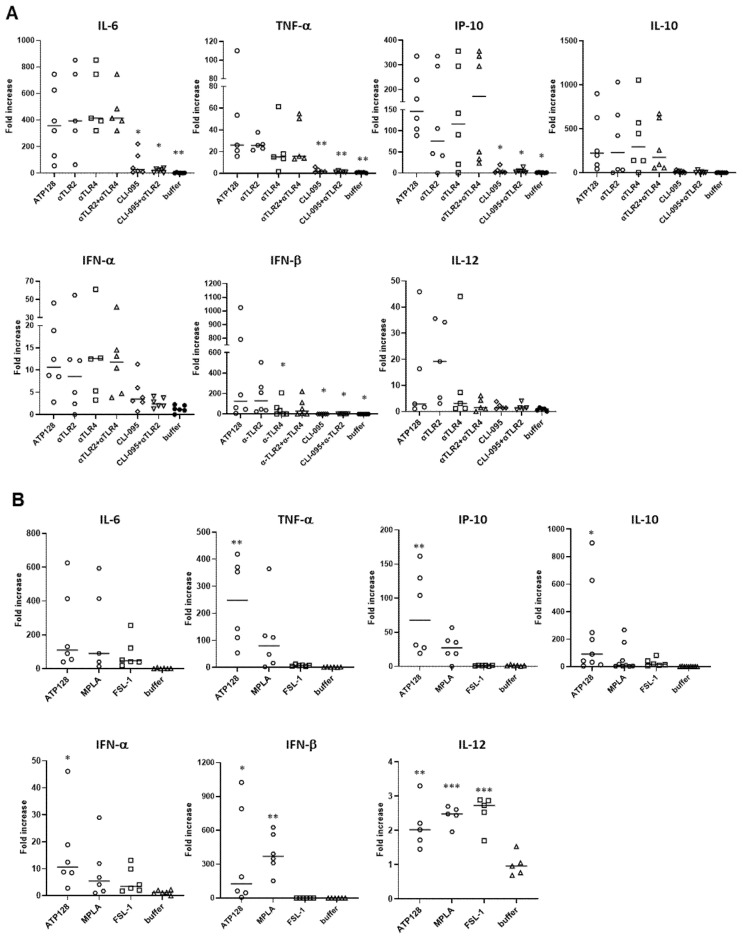
ATP128 induces cytokines release upon activation of both the NF-κB and IRF3 pathways. (**A**) Human moDC were pre-treated with anti-TLR2, anti-TLR4, or CLI-095 for 1 h, then stimulated for 6 h or 24 h (for IL-12) with 300 nM ATP128. Cell supernatants were collected to assess the signature of cytokines secretion upon ATP128 stimulation through a multiplex cytokine assay. The values in the graphs represent the fold increase of the stimulated samples to the medium. Each shape is representative of a different buffy coat. Represented are the cytokines that are mainly induced upon stimulation. Values were compared via one-way ANOVA test to the sample stimulated with ATP128 FL. (**B**) Human moDC were incubated for 6 h or 24 h (for IL-12) with 300 nM ATP128 or 600 nM MPLA or 1 ng/mL FSL-1. Cell supernatants were collected to assess the signature of cytokines secretion through a multiplex cytokine assay (same as above). The values in the graphs represent the fold increase of the stimulated samples to the medium. Each shape is representative of a different buffy coat. Represented are the cytokines that are mainly induced upon stimulation. Values were compared via one-way Anova test to the sample treated with the buffer (negative control *** *p*-value < 0.001, ** *p*-value < 0.01, * *p*-value < 0.05.

**Figure 7 cancers-14-05134-f007:**
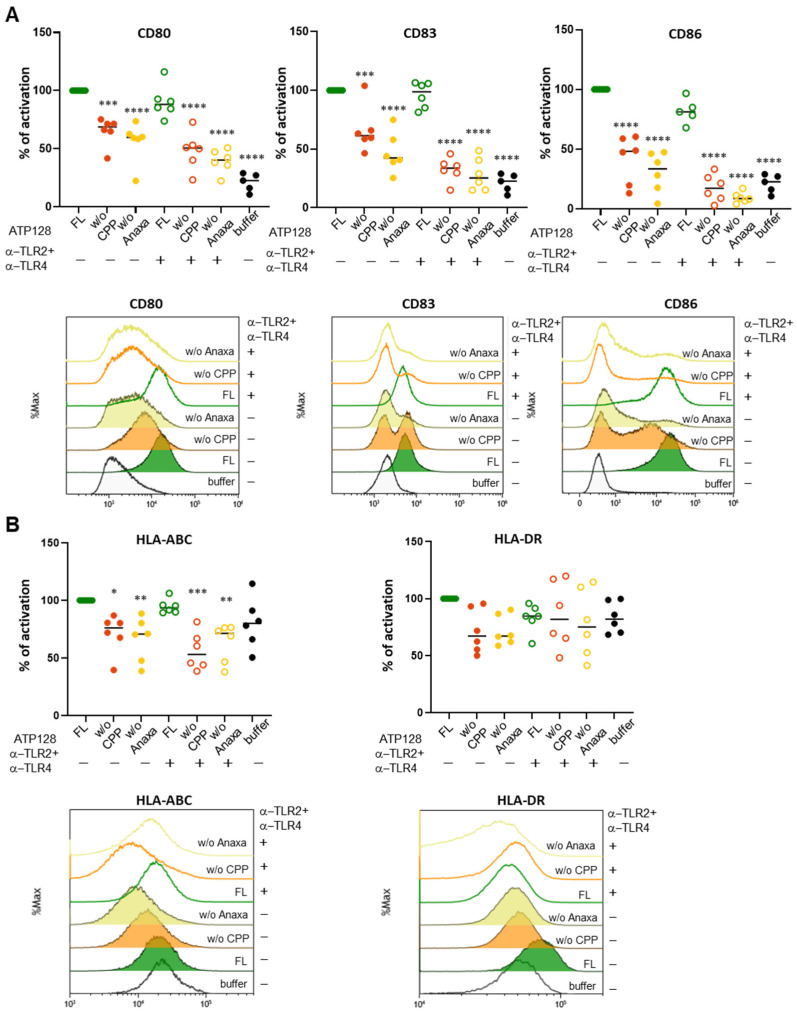
Both CPP and Anaxa are essential for human moDC activation. (**A**) Human moDC were pre-treated with a combination of anti-TLR2 and anti-TLR4 for 1 h, then stimulated for 24 h with 300 nM ATP128 FL, ATP128 *w*/*o* CPP, or ATP128 *w*/*o* Anaxa. Cells were collected after stimulation and surface costimulatory molecules (**A**) and HLA class I and class II molecules (**B**) were analyzed through flow cytometry. Each shape is representative of a different buffy coat. Histograms of one representative buffy coat are shown below. Values were compared via one-way ANOVA test to the sample stimulated with ATP128 FL. **** *p*-value < 0.0001, *** *p*-value < 0.001, ** *p*-value < 0.01, * *p*-value < 0.05.

**Figure 8 cancers-14-05134-f008:**
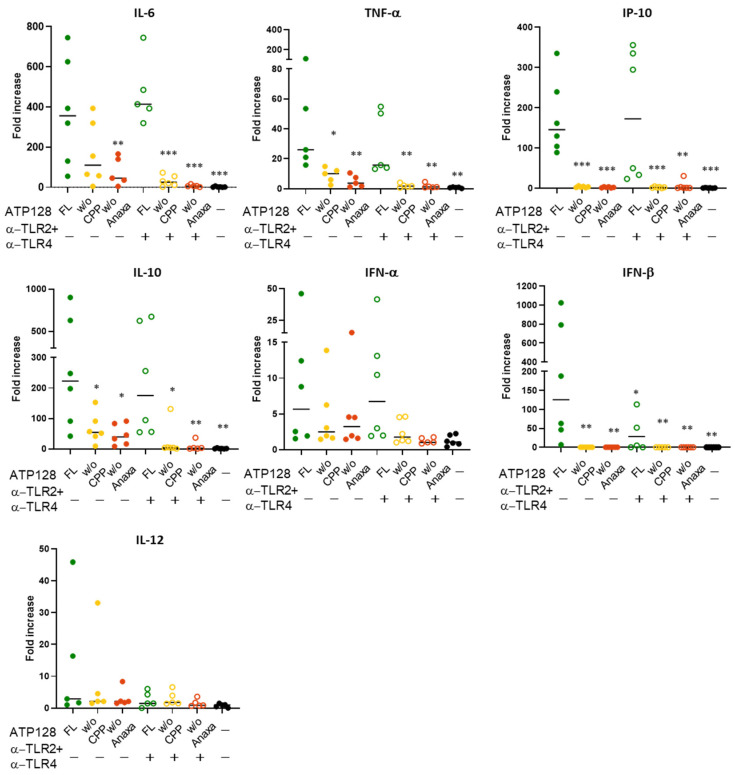
The absence of the CPP and Anaxa domain greatly affects the secretion of pro-inflammatory cytokines and type I IFN. Human moDC were treated with a combination of anti-TLR2 and anti-TLR4 and incubated for 6 h with 300 nM ATP128 FL, *w*/*o* CPP, or *w*/*o* Anaxa. Cell supernatants were collected to assess the signature of cytokines secretion upon ATP128 stimulation through a multiplex cytokine assay. The values in the graphs represent the fold increase of the stimulated samples to the medium. Each shape is representative of a different buffy coat. Represented are the cytokines that are mainly induced upon stimulation. Values were compared via one-way ANOVA test to the sample stimulated with ATP128 FL*** *p*-value < 0.001, ** *p*-value < 0.01, * *p*-value < 0.05.

## Data Availability

The data presented in this study are available in this article and Appendix A.

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
