# Peer review of "ATP128 Clinical Therapeutic Cancer Vaccine Activates NF-κB and IRF3 Pathways through TLR4 and TLR2 in Human Monocytes and Dendritic Cells"

_cancers, 2022, doi:10.3390/cancers14205134_

Round 1
Reviewer 1 Report
In the manuscript by Pascolutti et al., the authors analyze the effect of ATP128, a vaccine composed of three different parts, on THP-1 as well as monocyte-derived DCs. While the study gives interesting insights, it has some limitations. In several experiments only three donors are used. As variations between human donors are high, the numbers of donors should be increased at least to five to increase statistical power. Further, the authors should perform mixed leukocyte reactions with naïve T cells in order to analyze, whether the activation of the moDCs also result in T cell proliferation and especially in induction of IFNg+ TH1 cells and IFNg+ cytotoxic T cells. As the authors do not show data for IL-12 secretion, it is important to demonstrate that the moDCs are capable of inducing the desired T cell response.
· Line 289-291: As shown in Fig. 2C, the absence of CPP completely abrogates the remaining NF-B activation, thereby confirming the importance of this domain in absence of available receptors at the cell surface and within the endosomes.
o The authors should clarify this sentence. The construct without CPP does not inhibit any activation of NFKB itself. It works only in combination with TLR2 and TLR4 blockade.
· 3.4. ATP128 induces IRF3 phosphorylation with a delayed kinetics compared to NF-B:
o Was this experiment performed several times? If yes, the authors should perform statistical analysis for Figure 3. Otherwise, they should repeat the experiment.
· All figures with flow cytometry data with moDCs: The authors should show their gating strategy for moDCs as a supplementary figure.
· In Figure 4A, the blocking antibody against TLR4 reduces the activation of half of the cells in the histogram plots shown for CD80 and CD86. However, in the upper panel showing the ratio, no effect is seen for TLR4. How did the authors calculate the values shown in the upper panel of Figure 4A?
· Figure 4B: In both the upper and lower panel, it looks like there is an effect of blocking TLR2 and TLR4 for HLA-DR expression. The authors should repeat the experiment as three donors seem to be not enough for proper statistic, especially if using ANOVA.
· Figure S3: MoDCs seem to express the same level of TLR2 as THP-1 cells. How do the authors explain that there is no effect of blocking TLR2, while it was the major receptor for NFKB activation of THP-1 cells?
· Figure 6: Instead of a ratio to medium, the authors should provide the measured concentration of the cytokines. Then, interpretation of the results would be easier.
· Discussion, L 527-528: “This leads to an increased APC activation with subsequent improvement and amplification of tumor-specific T cell responses, an essential requirement for tumor cell recognition and killing”
o This sentence is misleading as no T cell responses are shown in the human setting
· While I understand that THP-1 cells might mainly signal via TLR2 as they rather express low level of TLR4, it is not clear to me, why TLR2 can not compensate in moDCs when TLR4 is inhibited. As ATP128 is able to activate TLR2, it should also work in moDCs if TLR4 is blocked. Did the authors check for coreceptors of TLR2 (e.g. TLR1 and TLR6)? In case moDCs are missing TLR1 or TLR6 and THP-1 cells are positive, this might explain why ATP128 is not able to activate TLR2 in moDCs
· For moDCs, there is a strong discrepancy between blocking antibodies and inhibition via CLI095. How do the authors explain this? Does the antibody not work? Is the used concentration sufficient?
Reviewer 2 Report
In their article “ATP128 clinical therapeutic cancer vaccine activates NF-kB and IRF3 pathways through TLR4 and TLR2 in human monocytes and dendritic cells“, Pascolutti et al describe the in vitro and ex vivo evaluation of a novel candidate vaccine from the KISIMA platform. The vaccine, ATP128, consists of a CPP, a multi-antigenic domain (Mad) and a TLRag (anaxa) section and builds on previous vaccine constructs Mad5-12 (ref 1). Also the ATP128 candidate is already being tested in a phase I/II clinical trial. The current work investigates the intracellular mechanisms of action of the vaccine as a whole and its building components.
The manuscript describes the results from experiments in THP-1 cells (in vitro) and monocyte derived DC (ex vivo). The experiments that are performed seem logic and the results are convincing. So the manuscript should deserve publication. Given the use of anti-PD-1 and VSV-GP128 in the ongoing clinical trial, one wonders the effect of especially Ezabenlimab on activation pathways in similar experiments. However, I imagine it is too much to ask.
The authors should check the manuscript, because the terms in vitro and ex vivo are being mixed erroneously: line 17 the THP-1 cell line is called an ex vivo model, while it is an in vitro model. On the contrary in line 18 and line 340 monocyte derived DC are called an in vitro system, while they are ex vivo derived. In line 239 of the results and lines 460 and 496 of the discussion the THP-1 cell line is correctly named an in vitro system. Please correct where needed.
Other remarks: TLRag and anaxa are both used as alternatives at multiple times, in text and figures, which makes reading confusing. Use only one after defining.
In figure 4, 5 and 7, the y-axis is labelled ratio to ATP128 (%). Since a percentage is given, I don’t think this is a ratio, but the real percentage of a stimulation with ATP128 full length being 100% and inhibiting Ab or stimulation by other compounds or by defective constructs as percentage thereof. So that is not a ratio.
Round 2
Reviewer 1 Report
The authors addressed most of my concerns in their revised version of the manuscript. However some points should be edited:
Line 321-324:
“In addition, two experiments with different kinetics were performed (respectively 0, 30 min, 1h, 2h and 4h or 0, 1h, 2h and 4h), indicating a statistical difference in the percentage of IRF3 phosphorylation at 2h (data not shown)”.
Either the authors show the data in the manuscript or they should remove the statement about statistical difference. Further, it is not clear to me how they produced four data points with three experiments.
To Figure S4:
Did the authors use any further marker to analyze the moDCs (CD1a, CD11c, HLA-DR, CD14)? As moDCs frequently contain monocyte-derived macrophages, the purity of the cells should be checked and shown.
To Figure 6:
Usually, moDCs kept in medium do not produce measurable amounts of cytokines (i.e. above the level of detection of the assay) such as IL-12. In this case, it is not possible to calculate a fold increase as it is not possible to divide by 0. How did the authors account for that?
